# Protective humoral and cellular immune responses to SARS-CoV-2 persist up to 1 year after recovery

Chengqian Feng[1,5], Jingrong Shi[1,5], Qinghong Fan[1,5], Yaping Wang[1,5], Huang Huang[1,5], Fengjuan Chen[1,5], Guofang Tang[1], Youxia Li[1], Pingchao Li[2], Jiaojiao Li[1], Jianping Cui[1], Liliangzi Guo[1,3], Sisi Chen[1,3], Mengling Jiang[1], Liqiang Feng [2], Ling Chen [1,2], Chunliang Lei[1], Changwen Ke[4], Xilong Deng[1,6✉], Fengyu Hu[1,6✉], Xiaoping Tang[1,6✉] & Feng Li [1,6✉]

SARS-CoV-2 vaccination has been launched worldwide to build effective population-level immunity to curb the spread of this virus. The effectiveness and duration of protective immunity is a critical factor for public health. Here, we report the kinetics of the SARS-CoV-2 specific immune response in 204 individuals up to 1-year after recovery from COVID-19. RBD-IgG and full-length spike-IgG concentrations and serum neutralizing capacity decreases during the first 6-months, but is maintained stably up to 1-year after hospital discharge. Even individuals who had generated high IgG levels during early convalescent stages had IgG levels that had decreased to a similar level one year later. Notably, the RBD-IgG level positively correlates with serum neutralizing capacity, suggesting the representative role of RBD-IgG in predicting serum protection. Moreover, viral-specific cellular immune protection, including spike and nucleoprotein specific, persisted between 6 months and 12 months. Altogether, our study supports the persistence of viral-specific protective immunity over 1 year.

[1] Guangzhou Eighth People's Hospital, Guangzhou Medical University, Guangzhou, China. [2] Guangzhou Institutes of Biomedicine and Health, Chinese Academy of Sciences, Guangzhou, China. [3] Department of Gastroenterology, The First Affiliated Hospital, Jinan University, Guangzhou, China. [4] Guangdong Provincial Center for Disease Control and Prevention, Guangzhou, China. [5] These authors contributed equally: Chengqian Feng, Jingrong Shi, Qinghong Fan, Yaping Wang, Huang Huang, Fengjuan Chen. [6] These authors jointly supervised this work: Xilong Deng, Fengyu Hu, Xiaoping Tang, Feng Li. ✉email: gz8hdxl@126.com; gz8hhfy@126.com; tangxiaopinggz@163.com; gz8h_lifeng@126.com

Severe acute respiratory syndrome coronavirus 2 viruses (SARS-CoV-2) specific immune responses are critical for suppressing virus replication, ameliorating disease severity, and preventing reinfection[1–4]. Individuals recovered from wild-type SARS-CoV-2 infection could cross-protect from reinfection by a D614G mutant[1]. Deng et al. demonstrated that nonhuman primates exposed to SARS-CoV-2 develop potent antibody responses in the rhesus macaque infection model and are mostly immune to reinfection[2]. In a clinical trial testing the protective effect of neutralizing antibodies, outpatients who received neutralizing antibody therapy were less likely to have a COVID-19–related hospitalization or visit to an emergency department and were less likely to develop severe symptoms[3]. A two-dose regimen of BNT162b2 elicited robust immune protection against COVID-19 in persons aged >16 years[4]. SARS-CoV-2 convalescent patients and vaccine recipients usually generate robust immune responses. Protective immune responses against SARS-CoV-2 infection consists of two main parts, humoral immunoglobin (IgA and IgG) and T cell responses. The ideal aim is to achieve sterile infection; thus, most studies focus on protective serum IgG analysis for its ready availability[1,2,4,5]. A potent T cell immune protection will target viral containing cells, restrict viral spread in vivo, accelerate viral clearance, and alleviate disease burden. Viral-specific humoral and T cell responses can last, with a slight decline, up to 6 to 8 months in convalescent SARS-CoV-2 individuals[5,6]. Whether protective immune memory can be maintained longer term and whether antibodies or T cells can confer lasting protection requires further investigation, and understanding of this area will also benefit vaccine development.

Here, we analyze the dynamic changes of viral-specific humoral and cellular immune responses up to 1 year in a COVID-19 longitudinal follow-up cohort. In this representative cohort of patients, SARS-CoV-2 specific IgM and IgA levels decreased substantially to background levels within 1 year. Spike-specific IgG and serum protective capacity dropped significantly within the first 6 months but was maintained up to 1 year. SARS-CoV-2 specific T cell immune responses remained stable up to 1 year after hospital discharge. These findings imply that protective immune responses against SARS-CoV-2 can last at least 1 year.

## Results

**COVID-19 cohort characteristics.** A representative cohort of 204 recovered individuals (68.2%) out of the 299 patients over the same period who were admitted to Guangzhou Eighth People's Hospital, Guangzhou, China, from January 20 to February 29, 2020, participated in this follow-up study. Of these participants, 29 were diagnosed as having severe symptoms, 162 as moderate, 12 as mild, and one as asymptomatic (Table 1). Their ages and genders were representative of the whole patient population (Supp. Fig. 1). Except for the only asymptomatic patient, all individuals donated blood samples at multiple time points; fortunately, 50 individuals (24.5% of 204) accomplished the whole

round of follow-ups and contributed blood samples at all four follow-up time points. Thus, kinetics assessment of immune memory was possible. In short, the follow-up cohort represents the whole population of COVID-19 patients in Guangzhou at the beginning of 2020.

**Longitudinal changes of humoral antibodies against SARS-CoV-2.** The receptor-binding domain (RBD) of spike SARS-CoV-2 plays an essential role in viral binding to its receptor, angiotensin-converting enzyme 2 (ACE2), the levels of RBD-specific antibody correlates to and represents its neutralizing capacity[7]. We analyzed the kinetics of circulating RBD-specific IgM, IgA, and IgG antibodies over time with two-step indirect immunoassay electrochemiluminescence immunoassay kits[8]. First, all patients were positively seroconverted (at least one IgM >1 COI, IgA >1 COI, or IgG >1 COI) with at least one-time point, confirming an actual SARS-CoV-2 infection occurred for all the selected patients. Severe patients generated significantly higher RBD-IgA, RBD-IgG, and spike-IgG, but not RBD-IgM, than the moderate patients within the hospital stage (Fig. 1a–d and Supp. Fig. 2). Our result supported that more prolonged exposure of the immune system to viruses did enhance viral-specific antibody generation.

Next, we found that the level of RBD-specific IgM decreased dramatically from 4.669 (mean COI, hereinafter) to 0.785 (negative) (about sixfold, $p < 0.0001$, an unpaired $t$-test with Welch's correction) within 3 months after discharge (Fig. 1a). RBD-specific IgA also declined substantially from 9.283 to 3.234 (only 2.86-fold, $p < 0.0001$, an unpaired $t$-test with Welch's correction) within 3 months, but maintained up to 6 months (3.491, $p = 0.073$, an unpaired $t$-test with Welch's correction). At last, RBD-IgA became negative at 12-month point (0.57) ($p < 0.0001$, an unpaired $t$-test with Welch's correction) (Fig. 1b). Since IgA performs neutralization function in the mucosa, its serum levels were expected to be lower even at the peak than that in mucosa tissues. Whether mucosa tissues could maintain a stable IgA requires further investigation.

RBD-IgG was regarded as the critical indicator of immune protection against virus infection. Our analysis showed that RBD-IgG had a 2.87-fold reduction from 825 to 287 AU/ml within the first 3 months ($p < 0.0001$, an unpaired $t$-test with Welch's correction), and further to 153 AU/ml at the 6-month time points ($p = 0.0033$) (Fig. 1c, an unpaired $t$-test with Welch's correction), which was consistent with the previous reports[5,6]. Notably, the RBD-IgG levels maintained stably up to 1 year (170 AU/ml on average, Fig. 1c). The level of full-length spike-IgG, measured independently, exhibited a similar pattern to that of RBD-IgG (Fig. 1d), further corroborating the above observation that antibody would keep stable from 6 months to 1 year after a rapid decline.

RBD-IgG and full-length spike-IgG levels showed a strong positive correlation ($r = 7411$, $p < 0.0001$, $n = 474$, Pearson correlation, Fig. 1e). A strong positive correlation between RBD-IgG

**Table 1 Basic information of follow-up SARS-CoV-2 patients.**

| Groups | Follow-up /all patients (%) | In-patient stage | Follow-up stage | | | | With 4 points / All follow-up (%) |
|---|---|---|---|---|---|---|---|
| | | | 1-month | 3-month | 6-month | 12-month | |
| Severe | 29/55 (52.7%) | 23 | 20 | 19 | 13 | 12 | 8/29 (27.6%) |
| Moderate | 162/216 (75.0%) | 101 | 88 | 131 | 67 | 79 | 40/162 (24.7%) |
| Mild | 12/23 (52.2%) | 3 | 5 | 11 | 4 | 8 | 2/12 (16.7%) |
| Asymp. | 1/5 (20.0%) | 0 | 1 | 0 | 0 | 0 | 0/1 (0%) |
| total | 204/299 (68.2%) | 127 | 114 | 161 | 84 | 99 | 50/204 (24.5%) |

Frequency (of total) and percentage were shown. Asymp represents asymptomatic.

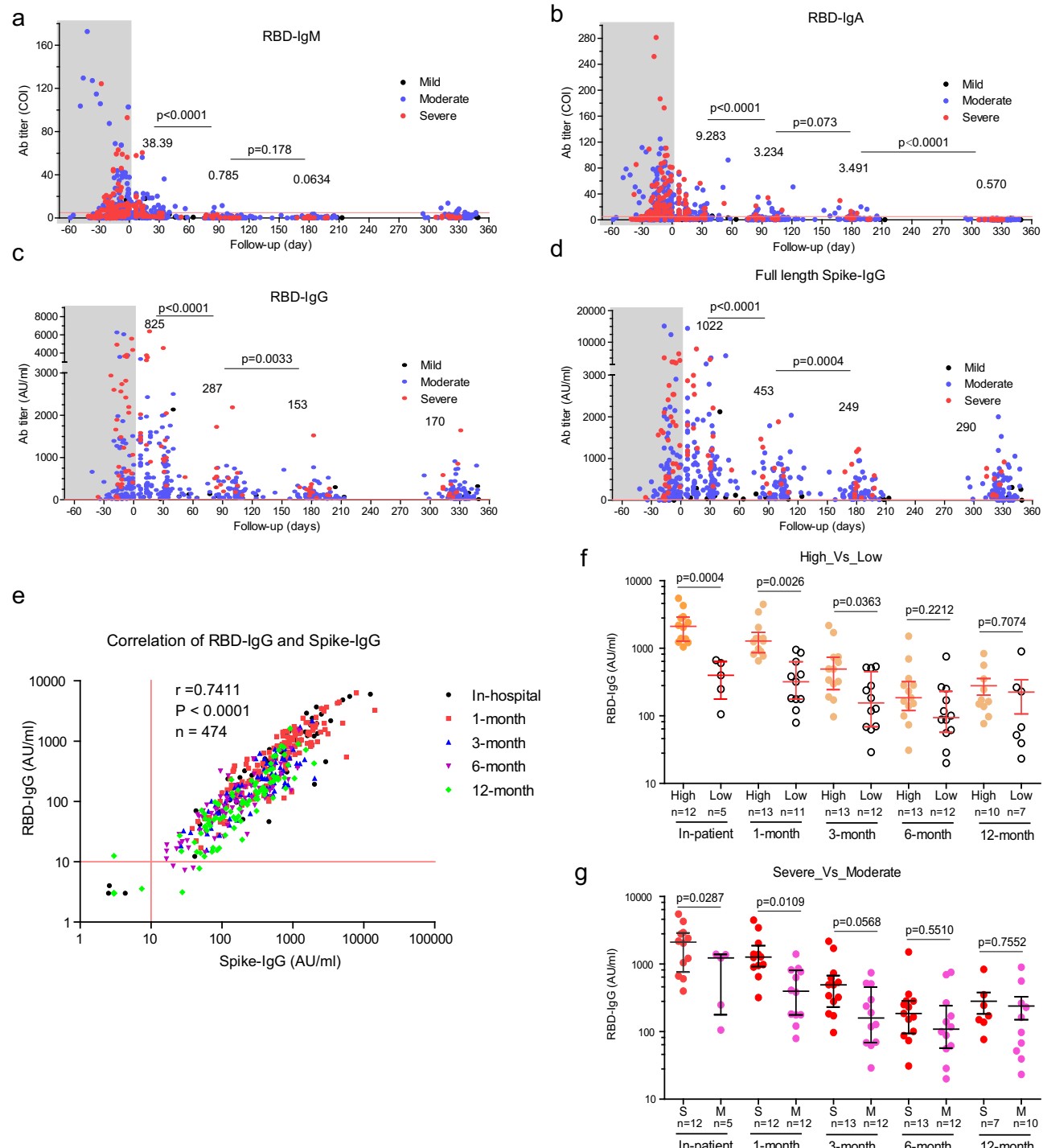

**Fig. 1 Kinetics of SARS-CoV-2 viral-specific antibodies up to 1 year.** SARS-CoV-2 infected individuals who attended the follow-up program were included in this analysis. **a–c** RBD-IgM, IgA, and IgG changes since symptom on-set. x-axis, follow-up time (day). y-axis, antibody titer (COI cut-off index; or arbitrary unit, AU/ml). The shaded regions represent the in-patient stage. The average values in the 1-month, 3-month, 6-month, and 12-month stages were labeled. P values (two-tailed) were shown. Mild (black dots), moderate (blue dots), and severe (red dots) symptoms. **d** Full-length spike-IgG changes since symptom on-set. x-axis, follow-up time (day). y-axis, antibody titer (AU/ml). **e** Correlation of RBD-IgG and Spike-IgG. In-hospital stage (black), 1-month follow-up (red), 3-month follow-up (blue), 6-month follow-up (purple), 12-month follow-up (green). Pearson r number and p value (two-tailed) were shown. **f** Longitudinal analysis of RBD-specific IgG between patients with high titer (RBD-IgG >1000 AU/ml, orange circle) and patients with low titer (RBD-IgG <1000 AU/ml, empty circle). **g** Longitudinal analysis of RBD-specific IgG between severe patients (red circle) and moderate patients (pink circle). Lines in x-axis and y-axis in **a–e** indicate the cut-off values. **a–d**, Mean values were labeled for each column. **f**, **g**, Data were presented as median with interquartile range. An unpaired t-test with Welch's correction analysis was applied, and p values (two-tailed) were labeled. Source data are provided as a Source Data file.

and full-length spike-IgG were observed in multiple time points (Fig.1e and Supp. Fig. 3). Thus, RBD-IgG was used in the following analysis. To investigate what kind of patients tends to maintain long-term RBD-IgG producing capacity, we test the influence of initial antibody titers on antibody persistence. We divided the follow-up individuals with at least the first three follow-up visits into high RBD-IgG (maximal titer >1000 AU/ml) and low (100 < maximal titer <1000 AU/ml) groups. We observed that the high group had significantly higher RBD-IgG levels at the in-hospital stage ($p = 0.0004$, an unpaired $t$-test with Welch's correction), 1-month ($p = 0.0026$, an unpaired $t$-test with Welch's correction), and 3-month ($p = 0.0363$, an unpaired $t$-test with Welch's correction) visits. But the difference diminished at the 6-month visit ($p = 0.2212$, an unpaired $t$-test with Welch's correction) and the 12-month visit ($p = 0.7074$) (Fig. 1f). Next, we demonstrated that disease severity does not necessarily contribute to higher RBD-IgG maintenance with individuals with at least three continuous follow-up serum samples (Fig. 1g). Therefore, our results showed that neither the antibody levels nor the disease severity would affect the RBD-specific IgG antibody persistence.

**Kinetics of serum protection against SARS-CoV-2.** We conducted a serum microneutralization assay using the alive SARS-CoV-2 infection system, the gold standard to evaluate serum protection against virus infection, to explore whether serum is still protective against SARS-CoV-2 infection. We found that the serum neutralizing titers decreased from 1-month to 3-month ($p = 0.0024$, an unpaired $t$-test with Welch's correction) and 6-month ($p = 0.0168$, an unpaired $t$-test with Welch's correction), in a similar manner to RBD-IgG and Spike-IgG (Fig. 2a). Subsequently, they kept stable until 12-month ($p = 0.591$, an unpaired $t$-test with Welch's correction), implying that SARS-CoV-2 specific humoral protection could persist long term. Since severe patients generated high levels of antibodies than moderate patients, whether the high levels could confer more potent protection was still unknown. We found that severe symptoms only resulted in a transiently higher neutralizing capacity at the earlier convalescence stage ($p = 0.020$, an unpaired $t$-test with Welch's correction), which diminished from the 3-month to 1-year visit (Supp. Fig. 4).

Then, we analyzed the correlation of RBD-IgG with the serum microneutralization. We found that a significant positive correlation existed between RBD-IgG and neutralization ($r = 0.4070$, $p < 0.0001$, $n = 475$, Pearson correlation) (Fig. 2b). Not surprisingly, the levels of Spike-IgG had a positive correlation with serum neutralization capacity (Supp. Fig. 4). Of note, the correlation between RBD-IgG and serum neutralization failed to reach statistical significance during the hospitalization stage ($r = 0.2101$, $p = 0.0905$, Pearson correlation, Supp. Fig. 5a) while positively maintained at 1-month ($r = 0.3952$, $p < 0.0001$, Pearson correlation), 3-month ($r = 0.2769$, $p = 0.0135$, Pearson correlation), 6-month ($r = 3846$, $p = 0.0005$, Pearson correlation), and 12-month ($r = 0.5216$, $p < 0.0001$, Pearson correlation) visits (Supp. Fig. 5b–e).

**Kinetics of SARS-CoV-2 specific T cell immune response.** Viral-specific memory T cells can eventually lessen the disease severity by eliminating virus-infected cells[9], forming another vital branch in combating virus infection when it occurs. To investigate whether protective T cellular immune response could persist, we generated Nucleoprotein (N) peptide pool because of its most abundant viral proteins[10] and Spike peptide pools because of their importance in viral infection and wide application[6]. Spike peptides were used to generate S1 and S2 peptide pools for convenient handling. We used the three peptide pools for gamma interferon (IFN-γ) enzyme-linked immunosorbent spot (ELISPOT) analysis of viral-specific T cells using fresh isolated PBMCs from the 6-month and 12-month visits. The ELISPOT detection results from one representative individual were shown (Fig. 3a). Compared to the convalescent patients in the hospital, total viral-specific IFN-γ secreting cells decreased at 6-month visits and kept stable to 12-month visits (Fig. 3b). Similar changes were observed for N, S1, and S2 proteins (Fig. 3c). Severe patients seem to generate persistent strong T cellular immune against Spike protein (Supp. Fig. 7). Composition analysis of the SARS-CoV-2 T cellular response revealed that the percentages of N, S1, and S2, are well maintained with only a slight decline (Fig. 3d), indicating the possible existence of long-lasting memory T just as the SARS infection in 2003[11].

## Discussion

In one aspect, our study demonstrated that both SARS-CoV-2 specific humoral immune responses can be maintained for up to 1 year. We observed that viral-specific RBD-IgG, Spike-IgG, and

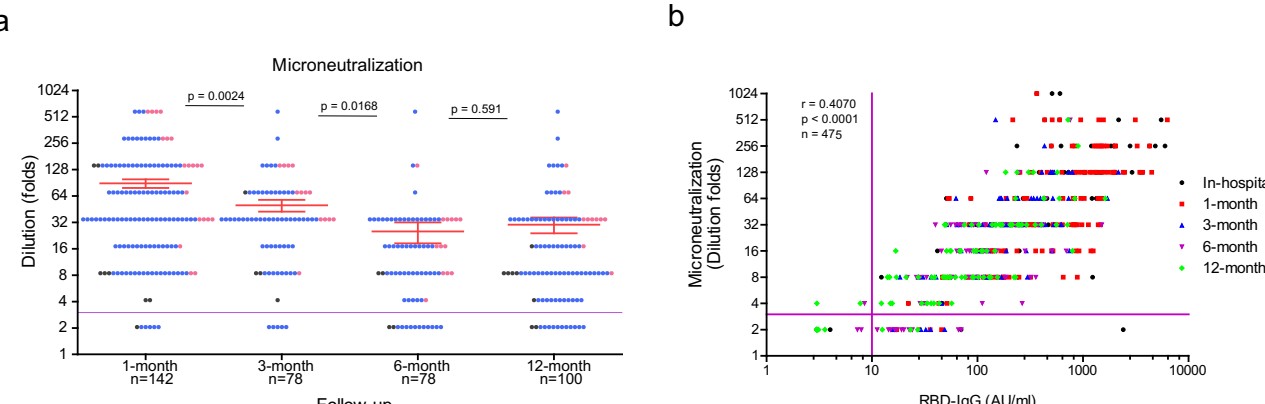

**Fig. 2 Longitudinal analysis of serum protection against SARS-CoV-2 infection.** Serum collected from multiple time points were tested against alive SARS-CoV-2 virus infection. **a** Microneutralization assay was performed using an alive SARS-CoV-2 infection cell culture system. y-axis, serum dilution folds. Data were presented as mean value ± SEM. Colored dots represent disease severity. Mild (blank dots), moderate (blue dots), and severe (red dots) symptoms. An unpaired $t$-test with Welch's correction analysis was applied, and $p$ values (two-tailed) were labeled. **b** Correlation of microneutralization (dilution folds) and RBD-IgG (arbitrary unit, AU/ml). A total of 475 pairs were analyzed, and Pearson $r$ and $p$ value (two-tailed) were shown. Lines in x-axis and y-axis in **a** and **b** indicate the cut-off values. Source data are provided as a Source Data file.

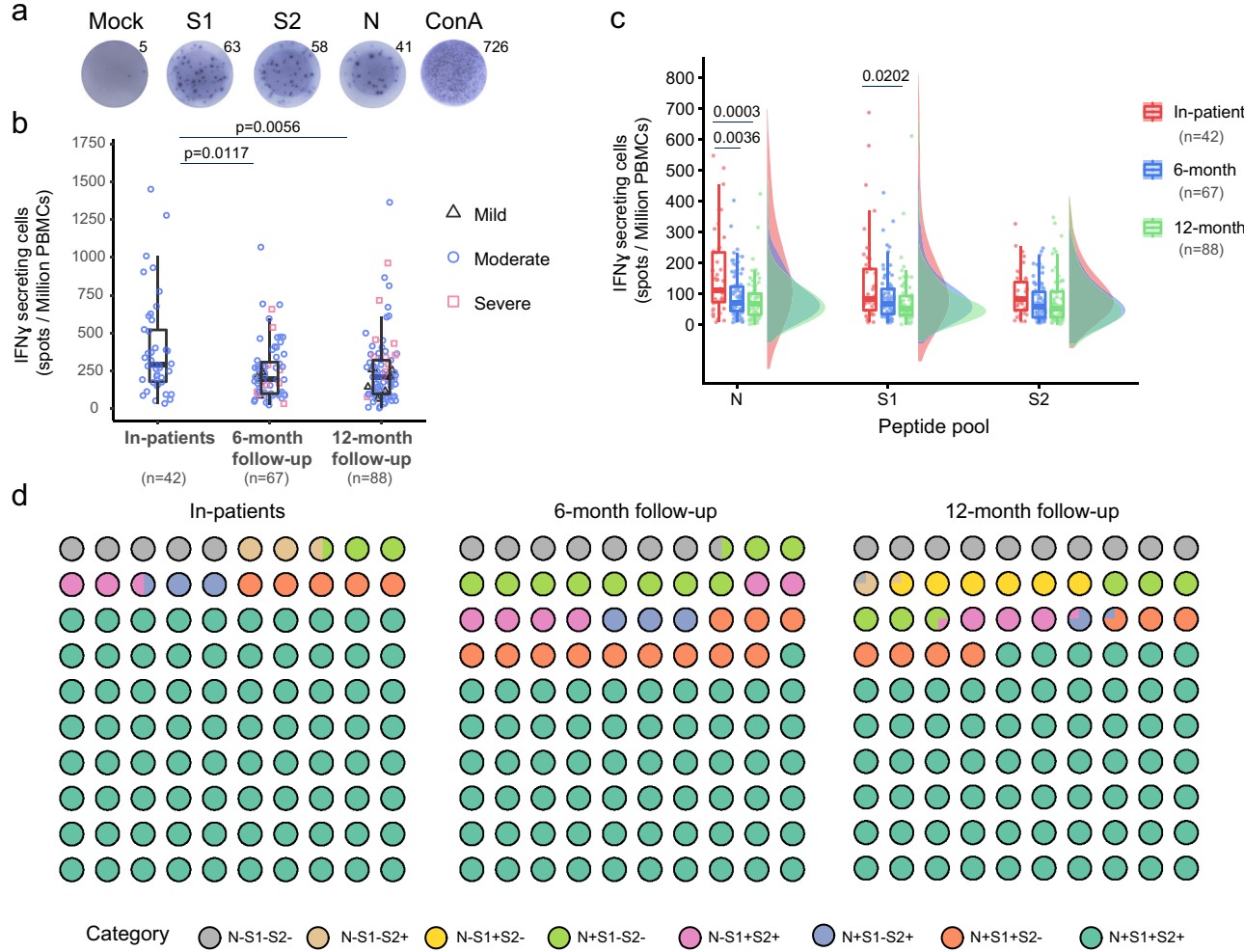

**Fig. 3 SARS-CoV-2 specific T cells response in recovered COVID-19 patients. a** Representative ELISpot well images displaying IFNγ-secreting cells. N, S1, and S2, Synthetic overlapping peptide pool of nucleocapsid (N), Spike 1 subunit (S1), and 2 (S2). Con A, Concanavalin A positive control. **b** Spot numbers of IFNγ-secreting cells indicating the reduction in SARS-CoV-2 specific immune response over time. The numbers are the sum of N, S1, S2-stimulated wells. Data were presented as median with interquartile range. **c** Raincloud plots show different T cellular response levels among three peptide pools at different time points of follow-up. **d** Waffle dot plot showing composition percentage of the SARS-CoV-2 response during 1-year follow-up. Statistics by Mann–Whitney *U*-test. *P* value (two-tailed) were shown. **b**, **c** Data were represented as box-and-whisker plots (median with interquartile, upper-limit, and lower-limit). Source data are provided as a Source Data file.

serum neutralization capacity declined quickly within 6 months, similar to the previous reports[5,6,12,13]. Fortunately, the three parameters keep stable after that to 1 year. It seems that the viral-specific IgG could keep the current levels for a while.

In the other aspect, our study showed that SARS-CoV-2 specific T cell response could also be relatively stable for the long term (Fig. 3). This observation is pretty crucial in the current situation. Multiple notorious strains[14–18], such as B.1.1.7 (also 501Y.V1, UK strain), B.1.351 (also 501Y.V2, Southern Africa strain), B.1.617.1 and B.1.617.2 (India strain), B.1.525 (also 20 A, Nigeria strain), and B.1.1.28.1(also 501Y.V3, Brazil strain) have emerged with either more infectiousness or pathogenesis or both. The novel mutants, mainly occurring in the RBD domain, could potentially escape the neutralization protection acquired either through natural infection or SARS-CoV-2 vaccination[15,17,18]. Unlike neutralizing antibodies, the viral-specific T cells could not block virus entry into the cells but still could kill the intracellular virus by directly destroying the virus-infected cells that present viral peptides[2,11]. As the numerous viral epitopes[19] can be produced from any viral proteins besides Spike protein for major histocompatibility complex presentation, novel mutations could

not easily break the T cell immune surveillance. In this perspective, vaccines that could induce durable viral-specific T cell immune responses will be promising to build an effective population immunity against SARS-CoV-2.

## Methods

**Patient information**. Guangzhou Eighth People's Hospital, as the official hospital appointed by the Guangzhou government, received and treated COVID-19 patients during the first wave of SARS-CoV-2 endemic spreading from January 2020 to March 2020. 299 patients admitted to Guangzhou Eighth People's Hospital were enrolled in this retrospective study. Two hundred and four patients participated in the follow-up study by providing at least once. General patient information, including age, sex, and clinical diagnosis, was collected from the hospital information system. All patients were diagnosed according to the criteria in the new Coronavirus pneumonia diagnosis and treatment plan (trial version 7) issued by the National Health and Health Commission. The severe symptom diagnosis was according to criteria as following:

1. Respiratory distress, respiratory rate (RR) ≥30 times/min in the resting state.
2. Oxygen saturation ≤93% in the resting state.
3. Arterial blood oxygen partial pressure (PaO2)/oxygen concentration (FiO2) ≤ 300 mmHg).
4. Deteriorated chest radiology imaging (X-Ray and high-resolution CT imaging) as an aid.

The moderate symptom was diagnosed when patients had visible pneumonia, and fever, and/or other respiratory symptoms. The Mild symptom was diagnosed when slight uncomforted but without pneumonia symptoms. Asymptomatic cases were diagnosed when the patients had no sign of clinical symptoms but were confirmed to be SARS-CoV-2 viral RNA positive during the follow-up stage after viral exposure or close contact with confirmed cases.

**Definitions.** All individuals participating in this follow-up program were divided into two stages: the in-patient and follow-up stages. The later was further grouped as 1-month (within 60 days after discharge), 3-month (70 to 135 days), 6-month (150 to 220 days), and 12-month (280 to 360 days).

**Measurement of viral-specific IgA, IgG, and IgM antibodies.** Plasma or serum samples were inactivated at 56 °C for 30 min and stored at −80 °C before testing. IgA, IgG, and IgM antibodies against the SARS-CoV-2 RBD spike protein in plasma samples were tested with two-step indirect immunoassay electrochemiluminescence immunoassay kits (Kangrun Biotech Co., Ltd.), according to the manufacturer's instructions. Briefly, the serum samples were first incubated with modified microparticles. The microparticles were coated with the RBD of the SARS-CoV-2 spike protein and with acridine ester-labeled antibodies against the Fc domain of human antibodies. After several rounds of washing off unbound substances, signal detection was performed on an automatic chemiluminescence immunoanalyzer (KAESER1000, Chongqing Cosmax Biotech Co., Ltd.). Similarly, IgG antibodies against the SARS-CoV-2 full-length Spike protein in plasma samples were tested with two-step indirect immunoassay electro-chemiluminescence immunoassay kits (Mairui Biotech Co., Ltd.), according to the manufacturer's instructions. All tests were performed under strict biosafety conditions. The antibody titer was tested once per serum sample. Antibody levels are presented as the measured chemiluminescence values divided by the cut-off (cut-off index, COI). COI <1 was regarded as negative, and COI >1 was regarded as positive.

**Microneutralization assay.** Heat-inactivated serum was serially diluted fourfold (from 1:4 to 1:1024) and then mixed with an equal volume (125 μl) of a viral solution containing 100 TCID50 of SARS-CoV-2. Next, serum-virus mixtures were first incubated for 2 h at 37 °C and were then applied to a semiconfluent VERO E6 monolayer in duplicate. After a 4-day incubation, virus-infected wells were assessed. Wells with visible cytopathogenic effect was defined as positive virus infection.

**Human IFN-γ ELISPOT assays.** IFN-γ ELISPO assays were performed as previously described (1) Briefly, MultiScreen 96-well filter plates (Merck Millipore, Darmstadt, Germany) were coated with antihuman IFN-γ monoclonal antibody (U-Cytech, Netherlands, No. CT640-10) overnight at 4° C. The wells were washed with PBS and blocked with R10 medium (RPMI-1640, 0.05 mM, 2-mercaptoethanol, 1 mM sodium pyruvate, 2mM L-glutamate, 10 mM 4-(2-hydroxyethyl)-1-piperazineethanesulfonic acid [HEPES], 10% fetal bovine serum [FBS], and penicillin/streptomycin (1×)) for 2 h at 37° C. Freshly isolated PBMCs were plated and stimulated in the presence of SARS-COV-2 S1 or S2 or N peptide pools (GenScript, China) for 24 h at 37° C. SARS-COV-2 peptides are 20 amino acids in length, with ten amino acids overlaps between sequential peptides. The plates were washed with PBS-Tween 20 (PBST), followed by incubation with biotinylated antihuman IFN-γ detection antibody (U-Cytech, Netherlands, No. CT640-10) and alkaline phosphatase (AKP)-conjugated streptavidin (BD, USA, No. 554065). Spots were developed by incubating in NBT/BCIP (Pierce, USA) for 10 min. Spots were counted using an ELISPOT reader (Bioreader 4000, BIOSYS, Germany). The number of spot-forming cells (SFC) is reported as the number of antigen-specific IFN-γ-secreting cells per million PBMCs.

**Statistical analysis.** The unpaired *t*-test with Welch's correction, Chi-square test, and Pearson correlation test were used to analyze the data in GraphPAD PRISM software (Version 5.01). Differences between groups were compared using the Mann–Whitney *U*-test and the Kruskal–Wallis rank-sum test for multiple comparisons with R studio (version 4.0.2).

**Ethical approval.** The study was approved by the medical ethics committee of Guangzhou Eighth People's Hospital (No. 202001134 and 202020153). Written consent was obtained from volunteer individuals.

**Reporting summary.** Further information on research design is available in the Nature Research Reporting Summary linked to this article.

## Data availability

The authors declare that all data that support the findings of this study are available within the paper and its supplementary information files. Source data are included as a Source Data file. Source data are provided with this paper.

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

## Acknowledgements

We thank all the members of the COVID-19 follow-up group for their indirect contributions to this study. We thank all the physicians and nurses who cared for these patients and collected the samples in the isolation ward. We also thank all the laboratory technicians who handled the samples and performed the measurements in Guangzhou Eighth People's Hospital. This work was supported by the National Natural Science Foundation of China (Nos. 81670536, 81770593, and 82061138006), National Grand Program on Key Infectious Disease Control (2017ZX10202203-004-002 and 2018ZX10301404-003-002), Guangdong Provincial Department of Science and Technology Fund (No. 2020B1111330002), Guangzhou Health Science and Technology Program (20211A010029), Sino-German Center for Research Promotion (SGC)'s Rapid Response Funding Call for Bilateral Collaborative Proposals Between China and Germany in COVID-19 Related Research (No. C-0032), and Emergency Key Program of Guangzhou Laboratory (EKPG21-29).

## Author contributions

X.D., F.H., C.L., X.T. and F.L. were responsible for the study design, study supervision, and data analysis. F.H. and F.L. wrote the paper. C.F., Y.W., H.H., F.C., G.T., Y.L., J.L., J.C. and M.J. were responsible for patient recruitment and sample and clinical data collection. J.S., P.L., L.F. and L.C. carried out the sequencing and bioinformatics analyses. Q.F., L.G. and S.C. conducted the viral RNA detection and antibody tests. C.K. performed the microneutralization assay.

## Competing interests

The authors declare no competing interests.
