## [Peer Review File · Nature Communications]

REVIEWER COMMENTS

Reviewer #1 (Remarks to the Author):

Thank you for the opportunity for review of the paper titled "Protective humoral and cellular responses against SARS-CoV-2 persist up to 1 year while RBD-specific IgG vanishes". The authors found that SARS-CoV-2 receptor-binding domain (RBD) specific immunoglobulin-M, A and G (IgM, IgA, and IgG) decreased over time and vanished at 12-month after discharge, serum neutralizing capacity and Spike and nucleoprotein (N) specific cellular persisted up to 12-month.

I have some concerns and suggestions for the authors.

1. The authors are welcome to provide study protocol. It is not clear whether this study has been registered online?
2. It is not clear how the cohort of long term follower were selected. It seems that different number of patients came back for follow-up study. The readers would like to know how many patients contributed blood samples at all time points.
3. In this study, can the authors provide the definition of severity, moderate and mild?
4. Figure 1: X-axis: Is correct that 0 means the day of discharge? The readers would be confused by -30 and -60 days.
5. Can the authors give some explanation of the remaining neutralizing capacity and cellular immunity against SARS-CoV-2?

Reviewer #2 (Remarks to the Author):

In the study by Feng et al, the serum antibody binding, neutralization and T cell responses were monitored in a cohort of donors who recovered from SARS-CoV-2 infection for up to 1 year. They report that the IgM IgA and IgG responses to the Receptor binding domain are high early in infection but decline by 1 year, although RBD-specific IgG titers are still detectable. In contrast neutralizing activity maintained, and RBD binding and neutralizing titers correlate. Curiously neutralizing titers do not decline as much by 1 year suggesting other epitopes outside the RBD are important for neutralization. In contrast to antibody responses the authors show that a population of cells (likely T cells but are they CD4 or CD8?) that secrete interferon gamma in response to peptide pools derived from spike or nucleocapsid are maintained for up to one year. As far as I know this is one of the farthest-out longitudinal studies of immune responses to SARS-CoV-2 infection. The results largely agree with other studies that look at antibody and T cell responses 6-8 months after infection which show they decline over time. I'm somewhat perplexed by the maintenance of neutralizing titers despite the lack of RBD responses as several studies have now shown the RBD is the major target of neutralizing antibodies.

One concern I have is that the commercial assay used to measure antibody binding to the RBD is not that sensitive. If so could potentially explain the discrepancy between RBD binding antibodies and neutralization.

To be clear, I think that since the same assay was used for all samples in this study the comparisons within samples are totally valid, but the sensitivity could still be low.

I would like to see an experiment where the authors deplete the RBD-specific antibodies from serum at the 1 year post infection time point and repeat the neutralization assays. If there is no change they could argue more strongly that the neutralizing epitopes targeted 1 year post-infection are not directed at the RBD.

It would also be prudent to look at binding responses to the full length spike. Presumably if there are neutralizing antibodies binding outside of the RBD, they should be measurable against the full length spike.

Reviewer #1 (Remarks to the Author):

Thank you for the opportunity for review of the paper titled “Protective humoral and cellular responses against SARS-CoV-2 persist up to 1 year while RBD-specific IgG vanishes”. The authors found that SARS-CoV-2 receptor-binding domain (RBD) specific immunoglobulin-M, A and G (IgM, IgA, and IgG) decreased over time and vanished at 12-month after discharge, serum neutralizing capacity and Spike and nucleoprotein (N) specific cellular persisted up to 12-month.

I have some concerns and suggestions for the authors.

1. The authors are welcome to provide study protocol. It is not clear whether this study has been registered online?

Response: We are happy to explain our study protocols. This study involved two related study protocols. The first one (No. 202001134), as stated in the Ethical approval part in the manuscript, was a generalized study protocol that covered the sample and information collection before negative viral conversion and patient discharge. After discharge, all patients were required to be isolated in the normal wards, like a hotel, for another two weeks for monitoring their possible virus recurrence. Meanwhile, they voluntarily entered into the second study protocol (No. 202020153), a follow-up study to monitor the possible long-term side effect on their health. Because as we have known, a large percentage of those patients survived the SARS infection in 2003 still suffered their side effect for pretty long time. They were followed up at one week, 1-month, 3-month, 6-month, and 12-month time points since their discharge. Their health conditions were systematically evaluated, especially on lung function.

We can see that the follow-up time was not strictly on-time, but over a

relatively wide window, in our study--for instance, the 3-month follow-up time covered from 70 to 135 days post-discharge. A series of factors affect their on-time follow-ups. 1) As all individuals were volunteers for the study, they chose to re-visit at their convenient time; 2) Patients, especially those severe patients, needed time to completely recover physically and emotionally; 3) Their outdoor activities were advocated to be maximally constrained to avoiding crowd in early 2020.

Unfortunately, our study protocols were only registered in our hospital but not online.

2. It is not clear how the cohort of long term follower were selected. It seems that different number of patients came back for follow-up study. The readers would like to know how many patients contributed blood samples at all time points.

Response: As we explained above, patients voluntarily participated in the follow-up study. They can choose their re-visits at their most convenient time. For example, they can skip the first three re-visits and choose the last time-point 12-month re-visit directly. In reality, some patients did.

We stated, “fortunately,50 individuals (24.5% of 204) accomplished the full round of follow-up”, in the original submission. To be more precise, we rephrased the sentence to be “fortunately, 50 individuals (24.5% of 204) accomplished the full round of follow-up **and contributed blood samples at all four time points**” in this revised version.

I should point out that it’s difficult to collect samples longitudinally from all individuals with their in-hospital stage and up to 1 year (with four follow-up time points). We can see that only 97 follow-up patients with the acute phase and 6-month post-discharge were included in the most comprehensive study recently reported by Wuhan Jinyintan Hospital in China (Lancet 2021, 397: 220–32)

We believe that adding your suggestion in the sentence will significantly facilitate the readers to get the critical information.

3. In this study, can the authors provide the definition of severity, moderate and mild?

Response: We have included the criteria for the definition of severity, moderate and mild in the method part as the following.

“All patients were diagnosed according to the criteria in the new Coronavirus pneumonia diagnosis and treatment plan (trial version 7) issued by the National Health and Health Commission. The severe symptom diagnosis was according to criteria as following: 1) Respiratory distress, RR \geq 30 times/min in the resting state; 2) Oxygen saturation \leq 93% in the resting state; 3) Arterial blood oxygen partial pressure (PaO₂) / oxygen concentration (FiO₂) \leq 300mmHg), and 4) deteriorated chest radiology imaging (X-Ray and high-resolution CT imaging) as an aid. Moderate symptom was diagnosed when patients had visible pneumonia, and fever, and/or other respiratory symptoms. Mild symptom was diagnosed when slight uncomforted but without pneumonia symptoms. Asymptomatic cases were diagnosed when the patients had no sign of clinical symptoms but were confirmed to be SARS-CoV-2 viral RNA positive during the follow-up stage after viral exposure, or close contact with confirmed cases.”

The criteria for severity definition vary among different countries and regions in their own COVID-19 patient managing system. Inclusion of our criteria will help the readers understand the association of disease severity with immune memory.

4. Figure 1: X-axis: Is correct that 0 means the day of discharge? The readers would be confused by -30 and -60 days.

Response: Yes, the 0 in the X-axis means the day of discharge. We have added one more sentence in the figure legend 1 to clarify its mean.

5. Can the authors give some explanation of the remaining neutralizing capacity and cellular immunity against SARS-CoV-2?

Response: We heartily thank your concerns about the discrepancy between RBD-IgG levels and neutralization capacity.

After we received the comments from the editor, we measured the RBD-IgG of serum samples from both 6-month and 12-month in an independent experiment. Disappointingly, we found that the RBD-IgG kit, used in the previous manuscript, could not produce consistent readout between different batches, resulting in underestimating RBD-IgG levels in the 12-month visit.

Alternatively, we employed one newly developed sensitive and reliable quantitative assay to measure the RBD-IgG levels instead of the previous assay. We used the same batch of reagents to quantify the serum RBD-IgG levels within three days to get consistent results.

To be more critical, we also assessed the changes of the full-length Spike-IgG as suggested and observed that Spike-IgG exhibited similar kinetics to RBD-IgG.

In addition, our RBD-IgG results get supported by different teams who are also working on the follow-up of recovered patients. Through private communications, we learn that their results are similar to ours that RBD-IgG declined within the first 6 months and keep stable after 6 months, further supporting our conclusion.

Therefore, we can confidently conclude that viral humoral immune protection still maintains in one year.

I hope you can be satisfied with our explanation.

Reviewer #2 (Remarks to the Author):

In the study by Feng et al, the serum antibody binding, neutralization and T cell responses

were monitored in a cohort of donors who recovered from SARS-CoV-2 infection for up to 1 year.

They report that the IgM IgA and IgG responses to the Receptor binding domain are high early in infection but decline by 1 year, although RBD-specific IgG titers are still detectable. In contrast neutralizing activity maintained, and RBD binding and neutralizing titers correlate. Curiously neutralizing titers do not decline as much by 1 year suggesting other epitopes outside the RBD are important for neutralization. In contrast to antibody responses the authors show that a population of cells (likely T cells but are they CD4 or CD8?) that secrete interferon gamma in response to peptide pools derived from spike or nucleocapsid are maintained for up to one year.

As far as I know this is one of the farthest-out longitudinal studies of immune responses to SARS-CoV-2 infection. The results largely agree with other studies that look at antibody and T cell responses 6-8 months after infection which show they decline over time.

I'm somewhat perplexed by the maintenance of neutralizing titers despite the lack of RBD responses as several studies have now shown the RBD is the major target of neutralizing antibodies. One concern I have is that the commercial assay used to measure antibody binding to the RBD is not that sensitive. If so could potentially explain the discrepancy between RBD binding antibodies and neutralization. To be clear, I think that since the same assay was used for all samples in this study the comparisons within samples are totally valid, but the sensitivity could still be low.

Response: We heartily thank your comment for the discrepancy between RBD-IgG levels and neutralization capacity.

After we received the comments from the editor, we measured the RBD-IgG of serum samples from both 6-month and 12-month in an independent experiment. Disappointingly, we found that the RBD-IgG kit, used in the previous manuscript, could not produce consistent readout between different batches, resulting in underestimating RBD-IgG levels in the 12-month visit.

Alternatively, we employed one newly developed sensitive and reliable quantitative assay to measure the RBD-IgG levels instead of the previous assay. We used the same batch of reagents to quantify the serum RBD-IgG levels within three days to get consistent results.

To be more critical, we also assessed the changes of the full-length Spike-IgG as suggested and observed that Spike-IgG exhibited similar kinetics to RBD-IgG.

In addition, our RBD-IgG results get supported by different teams who are also working on the follow-up of recovered patients. Through private communications, we learn that their results are similar to ours that RBD-IgG declined within the first 6 months and keep stable after 6 months, further supporting our conclusion.

Therefore, we can confidently conclude that viral humoral immune protection still maintains in one year.

Lastly, we appreciate your kind suggestion that we should include a Spike-IgG measurement.

I would like to see an experiment where the authors deplete the RBD-specific antibodies from serum at the 1 year post infection time point and repeat the neutralization assays. If there is no change they could argue more strongly that the neutralizing epitopes targeted 1 year post-infection are not directed at the RBD.

It would also be prudent to look at binding responses to the full length spike. Presumably if there are neutralizing antibodies binding outside of the RBD, they should be measurable

against the full length spike.

Response: First of all, we are sorry that we have no resource and facility to do a viral protection assay with serum depleted RBD-specific antibodies because of the exclusively limited times of BSL-3 labs available in Guangzhou, China.

Secondly, we replace the incorrect result about RBD-IgG levels over time after using a reliable and reproducible kit.

Finally, we included the full-length spike-specific IgG measurement in the new manuscript.

REVIEWERS' COMMENTS

Reviewer #1 (Remarks to the Author):

I am happy with the response of the authors to my concerns. The revised manuscript is much improved.

Reviewer #2 (Remarks to the Author):

The additional experiments that were included in the revised manuscript make much more sense, in the context of the RBD binding titers correlating with spike binding titers and with microneutralization titers. These results are consistent with newly published studies examining antibody responses ~ 1 year post infection (<https://doi.org/10.1038/s41586-021-03647-4>).

One of the statements in the discussion needs to be removed or revised.

"..., relieving the concern that immune protection will wane quickly and restoring our confidence in the persistence of population immunity against SARS-CoV-2 either acquired through natural infection or active immunization." Since the study examined infected rather than vaccinated individuals, there is no evidence presented here that vaccine elicited antibodies show comparable durability. A similar statement in the abstract should be edited or removed as well: "Altogether, our study suggested that vaccination elicited viral-specific immune protection could be durable after a rapid decline at the early period."

REVIEWERS' COMMENTS

Reviewer #1 (Remarks to the Author):

I am happy with the response of the authors to my concerns. The revised manuscript is much improved.

Response: We are very happy to hear from you that all concerns are addressed . Thanks!

Reviewer #2 (Remarks to the Author):

The additional experiments that were included in the revised manuscript make much more sense, in the context of the RBD binding titers correlating with spike binding titers and with microneutralization titers. These results are consistent with newly published studies examining antibody responses ~ 1 year post infection (<https://doi.org/10.1038/s41586-021-03647-4>).

One of the statements in the discussion needs to be removed or revised.

"..., relieving the concern that immune protection will wane quickly and restoring our confidence in the persistence of population immunity against SARS-CoV-2 either acquired through natural infection or active immunization." Since the study examined infected rather than vaccinated individuals, there is no evidence presented here that vaccine elicited antibodies show comparable durability. A similar statement in the abstract should be edited or removed as well: "Altogether, our study suggested that vaccination elicited viral-specific immune protection could be durable after a rapid decline at the early period."

Response: We heartily thank your suggestions to improve our original version of manuscript. Now, we are very happy to know all the major concerns have been addressed.

According to your comments, we have deleted the statement in the abstract (Page 4).

~~Altogether, our study suggested that vaccination elicited viral specific immune protection could be durable after a rapid decline at the early period.~~

We also have deleted the statement in the discussion part (page 8).

~~, relieving the concern that immune protection will wane quickly and restoring our confidence in the persistence of population immunity against SARS-CoV-2 either acquired through natural infection or active immunization~~